# Muscle Pathology in Dystrophic Rats and Zebrafish Is Unresponsive to Taurine Treatment, Compared to the *mdx* Mouse Model for Duchenne Muscular Dystrophy

**DOI:** 10.3390/metabo13020232

**Published:** 2023-02-04

**Authors:** Jessica R. Terrill, Corinne Huchet, Caroline Le Guiner, Aude Lafoux, Dorian Caudal, Ankita Tulangekar, Robert J. Bryson-Richardson, Tamar E. Sztal, Miranda D. Grounds, Peter G. Arthur

**Affiliations:** 1School of Molecular Sciences, The University of Western Australia, Perth 6009, Australia; 2TaRGeT Lab, Translational Research for Gene Therapy, INSERM, UMR 1089, Nantes Université, CHU Nantes, 440200 Nantes, France; 3Therassay Platform, CAPACITES, Nantes Université, 44007 Nantes, France; 4School of Biological Sciences, Monash University, Melbourne 3800, Australia; 5School of Human Sciences, the University of Western Australia, Perth 6009, Australia

**Keywords:** duchenne muscular dystrophy, DMD^mdx^ rat, *dmd* zebrafish, taurine, therapies for DMD, animal models for DMD

## Abstract

Inflammation and oxidative stress are strongly implicated in the pathology of Duchenne muscular dystrophy (DMD), and the sulphur-containing amino acid taurine ameliorates both and decreases dystropathology in the *mdx* mouse model for DMD. We therefore further tested taurine as a therapy using dystrophic DMD^mdx^ rats and *dmd* zebrafish models for DMD that have a more severe dystropathology. However, taurine treatment had little effect on the indices of dystropathology in both these models. While we and others have previously observed a deficiency in taurine in *mdx* mice, in the current study we show that the rat and zebrafish models had increased taurine content compared with wild-type, and taurine treatment did not increase muscle taurine levels. We therefore hypothesised that endogenous levels of taurine are a key determinate in potential taurine treatment efficacy. Because of this, we felt it important to measure taurine levels in DMD patient plasma samples and showed that in non-ambulant patients (but not in younger patients) there was a deficiency of taurine. These data suggest that taurine homeostasis varies greatly between species and may be influenced by age and disease progression. The potential for taurine to be an effective therapy may depend on such variables.

## 1. Introduction

Duchenne Muscular Dystrophy (DMD) is a lethal, X-chromosome-linked muscle disease affecting about 1 in 3500–6000 boys worldwide (Reviewed in [1,2,3]). DMD is characterised by severe skeletal muscle loss caused by mutations in the dystrophin gene, resulting in defects or the absence of functional dystrophin protein in muscle. Dystrophin deficiency increases susceptibility to sarcolemma damage after muscle contraction, leading to myofibre necrosis (myonecrosis) associated with inflammation and oxidative stress, and subsequent myogenesis and regeneration plus fibrosis [4,5,6,7,8]. Repeated cycles of widespread myonecrosis result in the loss of muscle function in DMD patients, with premature death often due to respiratory or cardiac failure occurring usually by the third or fourth decade of life (reviewed in [1,9,10]).

There is currently no cure or effective treatment for DMD; research into therapies for DMD has focused on strategies to replace the missing dystrophin protein and on drugs to protect the dystrophic muscles from necrosis and reduce the severe dystropatholgy [10]. These drugs include anti-inflammatory agents, antioxidants, and drugs that target calcium homeostasis and fibrosis [10]. Most DMD preclinical drug research has utilised the classic *mdx* mouse, which has a naturally occurring mutation in the dystrophin gene [11,12]. In this widely used *mdx* mouse model, dystropathology is relatively mild and varies across the lifespan with an acute onset and peak of myonecrosis between about 21 and 28 days; after this growth period myonecrosis is reduced by about 8–12 weeks of age and stabilises to a progressive low level [13,14,15,16,17].

Taurine is an amino acid with many functions in tissue including anti-inflammatory and antioxidant effects [18]; our laboratory and several others have tested taurine in the *mdx* mouse with much success [19,20,21,22,23,24,25,26,27,28]. These combined studies showed that taurine treatment has a wide range of benefits on *mdx* dystropathology, including decreased myonecrosis and improved muscle strength [19,20,21,22,23,24,25,26,27,28] and that *mdx* muscle and plasma is deficient in taurine, relative to normal control wild-type (WT) levels, particularly during this early growth period of active myonecrosis [19,20,27,29,30,31,32]. We have similarly shown a deficiency of plasma taurine in the golden retriever muscular dystrophy (GRMD) dog model for DMD, that has a much more severe phenotype that closely resembles the pronounced dystropathology of DMD patients [33]. The GRMD dog model also exhibits high levels of oxidative stress and inflammation and would be an ideal candidate for further preclinical trials of taurine to progress this research to clinical trials [33,34]. However, preclinical trials in GMRD dogs are very expensive, variability between animals is high, and colonies are limited. Therefore, in order to advance our preclinical studies testing the potential of taurine as a therapy for DMD, we examined two other models of DMD that are considered to have a more severe phenotype than the *mdx* mouse. The *dmd^pc2^* zebrafish contains a nonsense mutation in exon 32 of the *dmd* gene, rendering it non-functional: these fish display a severe dystrophic phenotype [35] and are a powerful research tool for large scale and rapid screening of therapeutic drugs for DMD [35,36,37]. Another model investigated was the DMD^mdx^ rat, which was developed using transcription activator-like effector nucleases to target exon 23 of the *dmd* gene [38,39]. Normal rats are about 10 times larger than mice with a similar lifespan and the DMD^mdx^ rats appear to have a more severe and progressive dystropathology (compared with *mdx* mice) that more closely resembles the DMD patients, with sustained muscle necrosis, more pronounced fibrosis and adipose tissue, loss of strength, earlier changes in heart and nerves, and a reduced lifespan [38,40,41,42].

We have proposed that cells of the immune system, particularly activated neutrophils, exacerbate dystropathology, specifically because hypochlorous acid (HOCl) a potent reactive oxygen species is generated from myeloperoxidase (MPO) produced by neutrophils [33], and that the antioxidant effects of taurine in *mdx* mice are a consequence of taurine ameliorating the production, or effects, of HOCl [8,20]. Therefore, one aim of this study in DMD^mdx^ rats was to establish whether taurine treatment ameliorated oxidative stress and reduced the severity of dystropathology in dystrophic rat muscle. Taurine was administered to young male DMD^mdx^ rats aged 4 weeks for 8 weeks. Unexpectedly, taurine levels were higher in untreated DMD^mdx^ rats in both plasma and muscle, compared with WT normal rats, and taurine treatment of DMD^mdx^ rats had no impact on the taurine content in muscles.

To clarify our observations of conflicting taurine effects in the dystrophic mouse and rat models, we also tested the impact of taurine treatment in dystrophic *dmd^pc2/pc2^* zebrafish. Similarly, *dmd^pc2/pc2^* zebrafish exhibited significantly higher taurine levels compared with WT controls. In addition, taurine treatment from 2 to 6 days post-fertilisation (dpf) increased taurine content in WT control fish but did not further increase taurine levels in *dmd^pc2/pc2^* zebrafish.

Since the ability of taurine to protect dystrophic muscle from damage is potentially dependent on intrinsic taurine levels, we felt it pertinent to establish the levels of taurine in DMD patients. Archived human plasma samples showed that plasma taurine levels were significantly lower in non-ambulant DMD boys (compared with healthy controls), in accordance with the data for dystrophic *mdx* mice, but in marked contrast with DMD^mdx^ rats and dystrophic zebrafish.

## 2. Materials and Methods

All reagents were obtained from Sigma-Aldrich Australia (Maquarie Park, NSW, Australia) unless specified.

### 2.1. Animals—Rats

Rat experiments were carried out in Nantes, France on dystrophic DMD^mdx^ rats [31,38] and normal control littermate wildtype (WT) (Sprague–Dawley) male rats. All animal experiments were conducted in strict accordance with the guidelines of the French National Research Council for the Care and Use of Laboratory Animals (Permit Number: APAFIS#10792–2017061316021120). Rats were maintained at the Institut de Recherche en Santé 2—Nantes University on a 12-h light/dark cycle, under standard conditions, with free access to food and drinking water. From 4 weeks of age, rats were randomly assigned to the following groups—untreated wildtype (WT) control rats, untreated DMD^mdx^ rats, taurine-treated DMD^mdx^ rats (2 g/kg/day) and taurine-treated DMD^mdx^ rats (5 g/kg/day). These doses were chosen based on doses used in previous studies in our laboratory and others [20,22,23]. Taurine was administered via the drinking water for 8 weeks; every two days taurine concentration was adjusted in water in order to correspond to each individual’s approximate daily water intake. During the experiment, body weights and taurine consumption were measured.

At 12 weeks of age at the end of the experiment (after 8 weeks of treatment), forelimb grip strength and locomotor and behavioural activity of all rats were measured. To measure grip strength, rats were placed with their forepaws on a T-bar, and gently pulled backwards until they released their grip. A grip meter (Bio-GT3, Bioseb, Vitrolles, France) attached to a force transducer measured the generated peak force. For each session, 5 trials were performed sequentially, and the results are expressed as the mean of 3 median values in grams (g) and normalised by the body weight (g/g). Locomotion and behavior analysis was performed in an open-field arena (dimensions of 100 cm × 100 cm × 40 cm) for 5 min (OF-3C, Bioseb, Vitrolles, France). This open-field system was composed of a 3D sensor-based technology accurately capable of rearing detection by direct height measurements. This 3D camera was connected to software analyzing the spontaneous locomotion of the rat placed in a novel environment. Several parameters were analyzed: distance travelled (cm), activity time (s), and number of rearings.

Thirty minutes to 6 h after a subcutaneous injection of buprenorphine for analgesia (Buprecare, 0.03 mg/mL, Axience, Pantin, France), rats were anesthetised with etomidate (Hypnomidate 2 mg/mL, Maphar, Casablanca, Morocco) and ketamine (Imalgene, 100 mg/mL, Merial, Ingelheim am Rhein, Germany). Rats were sacrificed by blood sampling from the renal vein. Blood was taken for hematology analysis (to quantify blood neutrophils and other blood parameters), and the remainder of the blood was centrifuged to prepare plasma samples that were stored frozen at −80 °C. Both biceps femoris muscles were taken, and either frozen in pre-cooled isopentane for histological analysis, or snap frozen in liquid nitrogen for biochemical analysis. These frozen samples of plasma and muscle were flown to Perth, Australia for analyses. The tibialis anterior, extensor digitorum longus, and soleus muscles were also removed, and weight recorded.

### 2.2. Animals—Zebrafish

Zebrafish experiments were carried out in Melbourne, Australia. Fish maintenance and handling were performed as per the standard operating procedures approved by the Monash Animal Research Precinct Ethics Committee 3, Monash University and maintained according to standard protocols [43]. The *dmd^PC2/+^* zebrafish parental strains [35] were in-crossed and their progeny were treated with either 1 mM Taurine (as per [44]) or water (vehicle-treated control) dissolved in embryo media (E3) or left in E3 without treatment, from 2 to 6 days post fertilisation (dpf). For locomotion assays, fish were plated in a randomised manner in 24-well plates at 5 dpf and their movement was recorded at 6 dpf [45]. Following this, fish were genotyped using a KASP genotyping protocol (LGC Genomics-UK), and the distance travelled was determined using Ethovision XT, (Noldus, Wageningen, The Netherlands). At 6 dpf heads were removed and used for KASP genotyping, the tails were then snap frozen and flown to Perth, Australia. For the analysis, tails were pooled by genotype.

### 2.3. Human Plasma Samples

Archival human plasma samples from DMD patients and normal boys were obtained from the Newcastle University Biobank (UK). Samples from two groups of DMD patients were used, 10 ambulant young boys (aged 2–8) and 7 non-ambulant (aged 16–20). Control samples were taken from normal healthy boys of various ages and used as age matched controls (aged 2–8 and 18–20, respectively). Frozen samples were flown to Perth, Australia for analysis.

### 2.4. HPLC Quantification of Taurine in Rat and Human Plasma, Rat Muscles, and Zebrafish Tails

Taurine was measured in the plasma and muscle using reverse phase high performance liquid chromatography (HPLC) as previously described [20]. In brief, plasma samples were precipitated by the addition of 20 times by weight of 5% trichloroacetic acid (TCA). Frozen muscle was crushed using a mortar and pestle under liquid nitrogen and homogenised in 100 times 5% TCA. Zebrafish tails were collected from four independent replicates (12 tails for each experimental condition). The sample for each replicate was split into two, to allow technical replication, and the mean of the technical replicates was used in the subsequent analysis. Samples were homogenised in 200 µL 5% TCA. After centrifugation, supernatants were removed and stored at −80 °C before analysis. Analytes were separated using HPLC with fluorescent detection, with pre-column derivatisation with o-phthalaldehyde (OPA) and 2-mercaptoethanol (2ME). OPA reacts rapidly with amino acids and sulfhydryl groups to yield intensely fluorescent derivatives, and 2ME, a reducing agent, prevents the OPA reagent from oxidising. An internal standard, O-phospho-DL-serine, dissolved in 5% TCA was added. Sodium borate was used to adjust the pH to 9. Samples were placed in an autosampler, which was maintained at 4 °C. Samples were mixed on a sample loop with a derivatising solution containing 20 mM OPA and 60 mM 2ME in 100 mM sodium borate, pH 10, for 30 s before injection onto the column. Separation was achieved with a C18 column (4 µm, 4.6 × 100 mm, Agilent, Santa Clara, CA, USA) using an Agilent 1260 Infinity HPLC system. Mobile phase A consisted of 50 mM potassium phosphate buffer, methanol, and tetrahydrofuran (94:3:3). Mobile phase B consisted of 90% methanol, with a gradient increase in B from 0 to 100%. Fluorescence was set at 360 nm and 455 nm for excitation and emission, respectively. The protein content of the muscle and zebrafish samples were quantified by solubilising the pellet in 0.5M sodium hydroxide, before incubation at 80 °C for 15 min. Once fully dissolved, protein concentrations of supernatants were quantified using a Bradford protein assay (Bio-Rad Australia, South Granville, NSW, Australia).

### 2.5. Plasma Creatine Kinase (CK)

Plasma CK reflects the leak of CK from muscles into the blood and is a classic systemic measure of damage and necrosis of dystrophic muscles [46]. CK levels were measured using a CK-NAC kit (Randox Laboratories, Crumlin, United Kingdom) and analysed kinetically using a BioTek Powerwave XS Spectrophotometer (Currumbin, QLD, Australia) using the KC4 (V34) program.

### 2.6. Quantification of Myeloperoxidase (MPO) in the Muscle as a Measure of Neutrophil Activity

Myeloperoxidase (MPO) is an enzyme secreted by neutrophils and MPO activity is a useful biomarker of neutrophils in tissues [47,48]. The enzyme MPO catalyses the production of HOCl from hydrogen peroxide and chloride [49] and HOCl acid reacts with 2-[6-(4-aminophenoxy)-3-oxo-3H-xanthen-9-yl]benzoic acid (APF) to form the highly fluorescent compound fluorescein, that is measured in this method, as previously described [33]. Briefly, frozen biceps femoris muscles from rats were crushed under liquid nitrogen and homogenised in 0.5% hexadecyltrimethylammonium bromide in phosphate buffered saline (PBS). Samples were centrifuged and supernatants diluted in PBS. Human MPO was used as the standard for the assay (Cayman Chemical, Ann Arbor, MI, USA). Aliquots of each experimental sample or MPO standard were pipetted into a 384 well plate, before the addition of APF working solution (20 µM APF and 20 µM hydrogen peroxide in PBS) was added. The plate was incubated at room temperature (protected from light) for 30 min, with the fluorescence being measured every minute using excitation at 485 nm and emission at 515–530 nm. The rate of change of fluorescence for each sample was compared to that of the standards and results were expressed per mg of protein, quantified using the DC protein assay (Bio-Rad Australia, South Granville, NSW, Australia).

### 2.7. Quantification of Muscle Oxidative Stress (Protein Thiol Oxidation)

Protein thiols are susceptible to oxidation by oxidants, leading to the reversible formation of disulphide bonds, and thus the percentage of thiols undergoing oxidation is a sensitive biomarker of oxidative stress [50]. Reduced and oxidised protein thiols were measured in rat biceps femoris muscles using the 2-tag technique as described previously [20]. In brief, frozen muscle was crushed under liquid nitrogen, before protein was extracted with 20% TCA/acetone. Protein was solubilised in an SDS buffer and protein thiols were labelled with the fluorescent dye BODIPY FL-N-(2-aminoethyl) maleimide (FLM, Invitrogen, Waltham, MA, USA). Following the removal of the unbound dye using cysteine, protein was re-solubilised in SDS buffer and oxidised thiols were reduced with tris(2-carboxyethyl)phosphine (TCEP) before the subsequent unlabelled reduced thiols were labelled with a second fluorescent dye Texas Red C2-maleimide (Texas red, Invitrogen, Waltham, MA, USA). The sample was washed in 100% TCA, followed by acetone, and resuspended in SDS buffer. Samples were read using a fluorescent plate reader (Fluostar Optima, BMG Labtech, Ortenberg, Germany) with wavelengths set at excitation 485 nm, emission 520 nm for FLM, and excitation 595 nm, emission 610 nm for Texas red. A standard curve for each dye was generated using ovalbumin and the results were expressed per mg of protein, quantified using the DC protein assay (Bio-Rad Australia, South Granville, NSW, Australia).

### 2.8. Statistics

For rat and human experiments, data were analysed using GraphPad Prism software (Boston, MA, USA). One-way ANOVAs and post-hoc (Tukey) testing to correct for multiple comparisons were used. For zebrafish assays, data were analyzed using a linear model in IBM SPSS Statistics (version 28) with replicate, genotype, and treatment as factors for the taurine analysis and replicate, genotype, treatment, and tracking system as factors for the swimming analysis. In both cases planned tests were conducted to explore genotype–treatment interactions. All results are presented as mean ± SEM with a significance level set at *p* < 0.05.

## 3. Results

### 3.1. Dystropathology (Grip Strength, Plasma CK, and Other Parameters) in WT Compared with Untreated and Taurine-Treated DMD^mdx^ Rats

At 12 weeks of age, the DMD^mdx^ rat muscle produced 30% less forelimb grip force than WT rats, and neither dose of taurine had any effect on grip strength (Figure 1A).

DMD^mdx^ rats had 11-fold more plasma CK than WT, with no effect of either dose of taurine (Figure 1B). Other measures of phenotype and dystropathology for WT and untreated and treated DMD^mdx^ rats are summarised in Table 1. Differences for untreated DMD^mdx^ rats (compared with WT) included significantly higher (1.1-fold) muscle weight for *tibialis anterior* muscle, and for plasma, (1.7-fold) higher levels of platelets, a higher (~3-fold) percentage of neutrophils, fewer (30%) lymphocytes, and a small increase in haematocrit. Neither dose of taurine had any effect on any parameter, except that both doses decreased the haematocrit back to WT levels.

### 3.2. Plasma and Muscle Taurine Content in WT Compared with Untreated and Taurine-Treated DMD^mdx^ Rats

Untreated DMD^mdx^ rats had 3.5-fold more plasma taurine than WT control rats (Figure 2A). The 2 g/kg/day and 5 g/kg/day doses of taurine increased DMD^mdx^ plasma taurine 4 and 7-fold, respectively (Figure 2A).

Skeletal muscles (biceps femoris) of untreated DMD^mdx^ rats had 2-fold more taurine, compared with WT (Figure 2B) with no effect of either dose of taurine (Figure 2B).

### 3.3. Inflammation and Oxidative Stress in the Plasma and Muscles of Taurine-Treated DMD^mdx^ Rats

DMD^mdx^ rats had 3.3-fold higher plasma neutrophils (%) than WT (Table 1, Figure 3A) and neither dose of taurine had any effect on plasma neutrophil levels (Figure 3A). Muscles (biceps femoris) of untreated DMD^mdx^ rats also had 2-fold more MPO than WT (Figure 3B) and the 2 g/kg/day dose of taurine decreased DMD^mdx^ muscle MPO by 80% (Figure 3B): in contrast the higher 5 g/kg/day taurine dose had no effect (Figure 3B). Protein thiol oxidation was 1.7-fold higher in muscles of DMD^mdx^ rats compared with WT (Figure 3C) and neither dose of taurine had any effect on muscle protein thiol oxidation (Figure 3C).

### 3.4. Muscle Function in Untreated and Taurine-Treated WT (dmd^+/+^), Heterozygous (dmd^+/pc2^) and Dystrophic Homozygous (dmd^pc2/pc2^) Zebrafish

We used locomotion assays on zebrafish at 6 dpf to determine if taurine treatment improved muscle function. Dystrophin-deficient *dmd^pc2/pc2^* zebrafish travelled a distance of 25% and 24% less than WT *dmd^+/+^* and *dmd^+/pc2^* fish, respectively (*p* < 0.01) (Figure 4). Taurine treatment had no significant effect on the swimming performance of any genotype (Figure 4).

### 3.5. Taurine Content in Untreated and Taurine-Treated WT (dmd^+/+^), Heterozygous (dmd^+/pc2^) and Dystrophic Homozygous (dmd^pc2/pc2^) Zebrafish

Zebrafish tails were used to determine taurine levels since they are comprised mostly of skeletal muscle. Dystrophin-deficient *dmd^pc2/pc2^* zebrafish tails had 1.7- and 1.5-fold higher taurine than WT *dmd^+/+^* and heterozygote *dmd^+/pc2^*, respectively (Figure 5). While taurine treatment had no effect on taurine levels of dystrophin-deficient fish tails, taurine treatment significantly increased the taurine content in *dmd^+/+^* (*p* < 0.001) and *dmd^+/pc2^* (*p* = 0.014) tails (Figure 5).

### 3.6. Taurine Content in Human Plasma of Normal Controls and Patients with DMD

There was no difference in plasma taurine content between young healthy controls (2–8 years) and ambulant DMD boys aged 2–8 (Figure 6). However, the older non-ambulant DMD patients (aged 16–20 years) had 30% lower plasma taurine than age-matched controls (aged 18–20 years). Non-ambulant DMD patients had 3-fold more plasma taurine than ambulant DMD boys (Figure 6). Also of note, plasma taurine in older controls in their late teens was 6-fold higher than for the younger controls aged under 10 years (Figure 6).

## 4. Discussion

We hypothesised that taurine treatment would reduce the severity of dystropathology in the DMD^mdx^ rat and *dmd* zebrafish, two models for DMD that are considered more severe compared to the widely used *mdx* mouse. Taurine is considered important for the function of skeletal muscle, where it modulates ion channel function, membrane stability, and calcium homeostasis, as well as having anti-inflammatory and antioxidant properties [51,52,53,54,55,56]. However, we showed that taurine administration to DMD^mdx^ rats had very little effect on the indices of dystropathology. Likewise, taurine administration to young dystrophic *dmd* zebrafish had no effect on muscle function (distance travelled). The surprising results for these two animal models contrast with our previous studies in young *mdx* mice (aged up to 6 weeks) where taurine had many benefits, with improved grip strength and decreased muscle necrosis, inflammation, oxidative stress, and plasma CK content [19,20,21]. Various benefits of taurine administration in *mdx* mice have also been widely reported by other groups for mice aged up to 12 months [22,23,24,25,26,27,28,57].

Interestingly, the endogenous plasma and muscle taurine levels in young adult DMD^mdx^ rats aged 12 weeks were unexpectedly higher relative to WT rats and, while both doses of taurine increased levels of plasma taurine in DMD^mdx^ rats, neither dose had an effect on muscle taurine levels. Likewise, taurine levels in dystrophic *dmd^pc2/pc2^* zebrafish tails (mainly composed of skeletal muscle) were higher than WT *dmd^+/+^* fish, and taurine treatment did not increase the taurine content of dystrophic *dmd^pc2/pc2^* tails (but did in WT *dmd^+/+^* fish). Taurine content of *mdx* muscle has been measured across the lifespan of *mdx* mice (summarised in Table 2). In most studies, muscle taurine levels in *mdx* muscle were either lower than, or equivalent, to WT (C57), and taurine was effective at reducing dystropathology. In one study (at 10 weeks), muscle taurine levels were observed to be higher than WT, and at this age, taurine was not effective at reducing dystropathology [28].

In these studies where taurine treatment was effective in reducing the severity of dystropathology, it was also observed that taurine treatment increased the taurine content of *mdx* muscle (summarised in Table 2). These previous results, combined with the new data in this current study, suggest that taurine is only efficacious in models where treatment improves the muscle content of taurine. This increase after exogenous taurine treatment was not observed in dystrophic models where intrinsic endogenous muscle taurine levels were high (i.e., DMD^mdx^ rats, *dmd^pc2/pc2^* zebrafish, and 10 week *mdx* mice [28]). Interestingly, in older mice (from 12 weeks) taurine in *mdx* plasma was higher than WT (summarised in Table 2). However, taurine treatment was still effective in older *mdx* mice, suggesting that the muscle content of taurine, rather than plasma content, is the important factor in determining efficacy. While the exact cause of perturbations in taurine levels in these models is not understood, we have previously explored taurine metabolism and transport in *mdx* mice [30]. We showed that taurine homeostasis, including synthesis in the liver, excretion in the kidney, and transport into the muscle was affected by dystrophy (and age). Future studies into taurine homeostasis in DMD^mdx^ rats, *dmd^pc2/pc2^* zebrafish, and even DMD patients would greatly help our understanding of the great variability in endogenous taurine levels in all of these species.

Since these combined results suggest that the ability of taurine to protect dystrophic muscle from damage is potentially determined by endogenous taurine levels, we thought it pertinent to measure taurine levels in DMD boys. As DMD muscle biopsies were not available, we measured taurine in archival plasma samples from DMD patients. While there was no significant difference in plasma taurine levels between young ambulant DMD (2–8 years) and normal boys of the same age, the older non-ambulant DMD boys with severe loss of muscle mass (16–20 years) had lower plasma taurine compared with normal boys. The observation that the normal teenage boys had markedly more taurine than younger normal boys, accords with previous reports that attributed this difference to the increased requirements of taurine during periods of peak growth [58]. Interestingly, in normal mice we observed that taurine levels were 5 and 10 times higher in the liver and plasma, respectively, in juvenile 18-day WT mice (during a period of peak growth just prior to weaning) compared with older WT mice [30]. These data support the need for high levels of taurine during normal growth. In *mdx* mice, this is also the age where we observed the biggest deficiencies in liver and plasma taurine in juvenile animals; this is also the time of peak myonecrosis [30] and the high metabolic demands of growth [59]. Benefits of taurine administration were pronounced in young *mdx* mice; however, this is in marked contrast with the situation in young zebrafish, presumably due to species differences. To further understand taurine metabolism in dystrophy, and to help assess the potential of taurine as a beneficial therapy for DMD, it would be informative to measure taurine levels in DMD muscles at different ages: unfortunately, this is difficult in DMD patients as it requires invasive muscle biopsies.

Finally, since the DMD^mdx^ rat is a relatively new model, we thought it pertinent to discuss the phenotypical observations of the model from this current study (Table 1). Previous research has suggested that the model has a severe progressive phenotype compared with *mdx* mice [38] and our new data for DMD^mdx^ rats show that at 12 weeks of age (young adult) dystropathology was evident as reduced grip strength and increased inflammation, oxidative stress, and plasma CK content compared with normal WT rats (although many other parameters measured were not affected). Studies of dystrophic sciatic nerves in DMD^mdx^ rats (aged 8 months) suggest that there is more pronounced ongoing myonecrosis, compared with *mdx* mice [41]. A level of sustained myonecrosis was also reported for another dystrophic rat model with no significant difference in the extent of muscle degeneration between week 4 and week 13; these authors also concluded that the pathological progression is more pronounced in dystrophic rats compared with classic *mdx* mice [60]. It would be useful to have a more detailed description of the extent of myonecrosis, regeneration, and fibrosis in a range of skeletal muscles (e.g., various limb muscles, diaphragm, tongue) across the lifespan for dystrophic DMD^mdx^ rats to compare with the wealth of information published for *mdx* mice.

In conclusion, taurine administration was not an effective treatment in young dystrophic DMD^mdx^ rats analysed at 12 weeks of age, nor in young *dmd^pc2/pc2^* zebrafish at 6 dpf. We hypothesise that this is due to the intrinsic high levels of taurine we observed in both models (compared to WT), and the resultant inability to therapeutically increase muscle taurine levels. We also showed that plasma taurine levels varied between young and older DMD patients. Limitations of this study include the specific ages of animals used, since pathology of dystrophic animals varies greatly across lifespan. Additionally, to understand the regulation in taurine levels in these different models, future studies could investigate taurine metabolism and transport, as both have been shown to be dysregulated in the *mdx* mouse [30]. Therefore, more research is required to better understand the potential of taurine to be used as a therapy for DMD boys.

## Figures and Tables

**Figure 1 metabolites-13-00232-f001:**
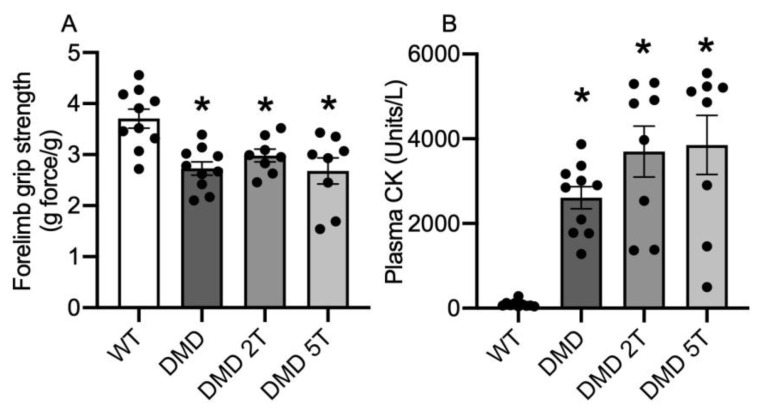
Markers of dystropathology, grip strength (**A**), and plasma CK (**B**), in WT and untreated and taurine-treated DMD^mdx^ rats (at 2 doses) aged 12 weeks. * = significantly (*p* < 0.05) different to WT. Bars represent mean ± SEM and n = 8–10 rats per group.

**Figure 2 metabolites-13-00232-f002:**
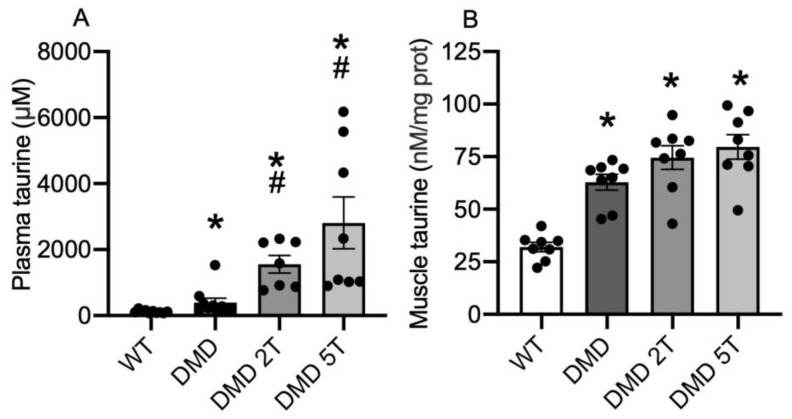
Levels of taurine in plasma (**A**) and muscle (**B**) in plasma, from WT and untreated and taurine-treated DMD^mdx^ rats (at 2 doses), aged 12 weeks. * = significantly (*p* < 0.05) different to WT. # = significant (*p* < 0.05) different to untreated DMD^mdx^ rats. Bars represent mean ± SEM and n = 8–10 per group.

**Figure 3 metabolites-13-00232-f003:**
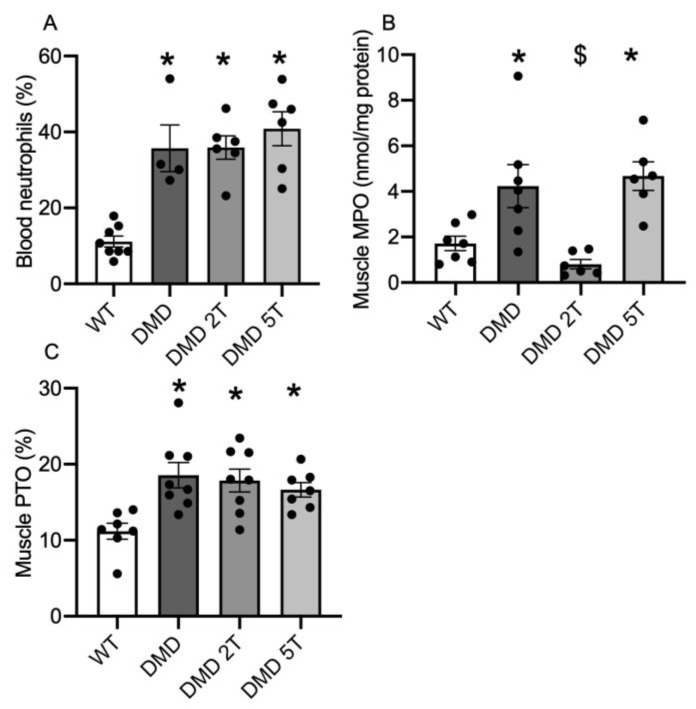
Markers of inflammation and oxidative stress in WT and untreated and taurine-treated DMD^mdx^ rats (at 2 doses) aged 12 weeks. Levels of plasma neutrophils (**A**) and muscle MPO (**B**) and oxidative stress measured as protein thiol oxidation (PTO) (**C**). * = significantly (*p* < 0.05) different to WT. $ = significantly (*p* < 0.05) different to untreated DMD^mdx^ rats. Bars represent mean ± SEM and n = 8–10 per group.

**Figure 4 metabolites-13-00232-f004:**
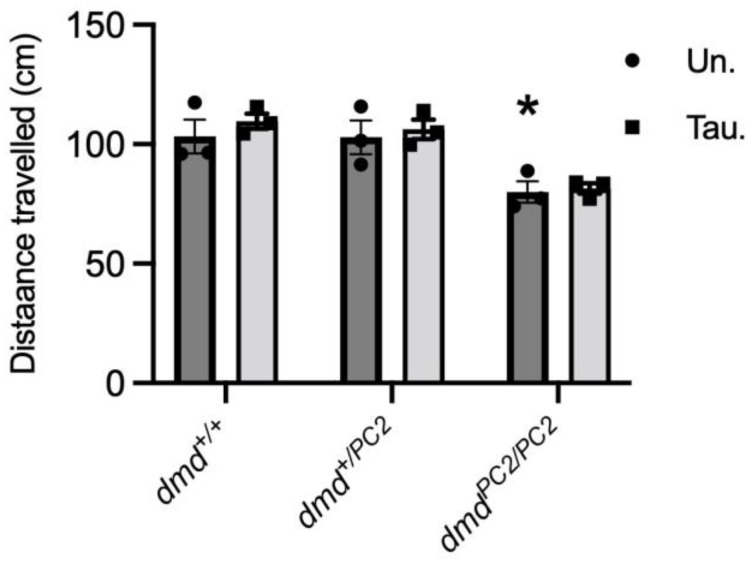
Muscle function (distance travelled) of 6 dpf WT (*dmd^+/+^*), heterozygous (*dmd^+/−^*) and homozygous (*^pc2/pc^*) dystrophic (*dmd^−/−^*) zebrafish treated with either 1 mM taurine or water (control) from 2 to 6 dpf. * = significantly (*p* < 0.05) different to WT. Bars represent mean ± SEM. For *dmd^+/+^*: n = 30, 27, 33 for untreated and n = 27, 22, 26 for taurine-treated; for *dmd^+/pc2^*: n = 48, 48, 55 for untreated and n = 56, 62, 55 for taurine-treated, and for *dmd^pc2/pc2^*: n = 41, 23, 28 for untreated and n = 32, 25, 37 for taurine-treated.

**Figure 5 metabolites-13-00232-f005:**
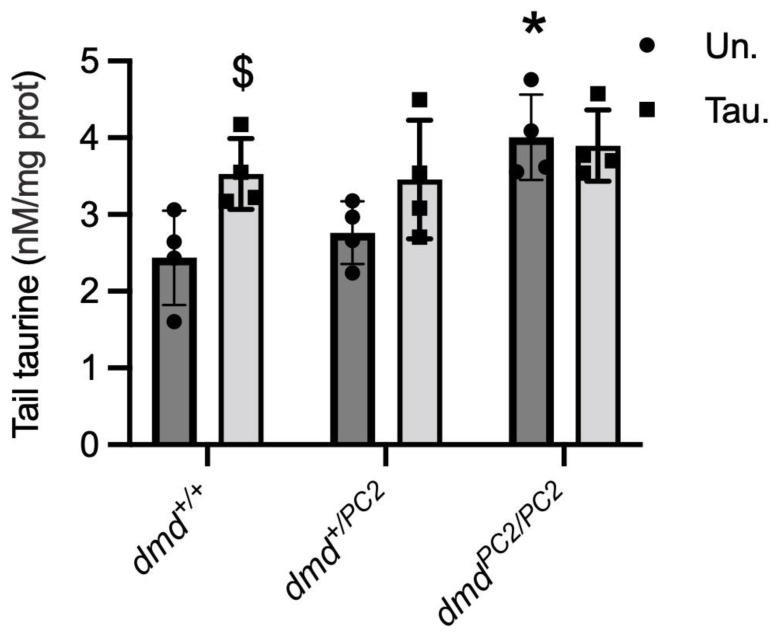
Levels of taurine in tails of WT (*dmd^+/+^*), heterozygous (*dmd^+/−^*) and homozygous (*^pc2/pc^*) dystrophic (*dmd^−/−^*) zebrafish treated with either 1 mM taurine (tau) or water (un) from 2 to 6 dpf. * = significantly (*p* < 0.05) different to WT. $ = significantly (*p* < 0.05) different to the untreated of same strain. Bars represent mean ± SEM and n = 4 (pooled samples of 12) per group.

**Figure 6 metabolites-13-00232-f006:**
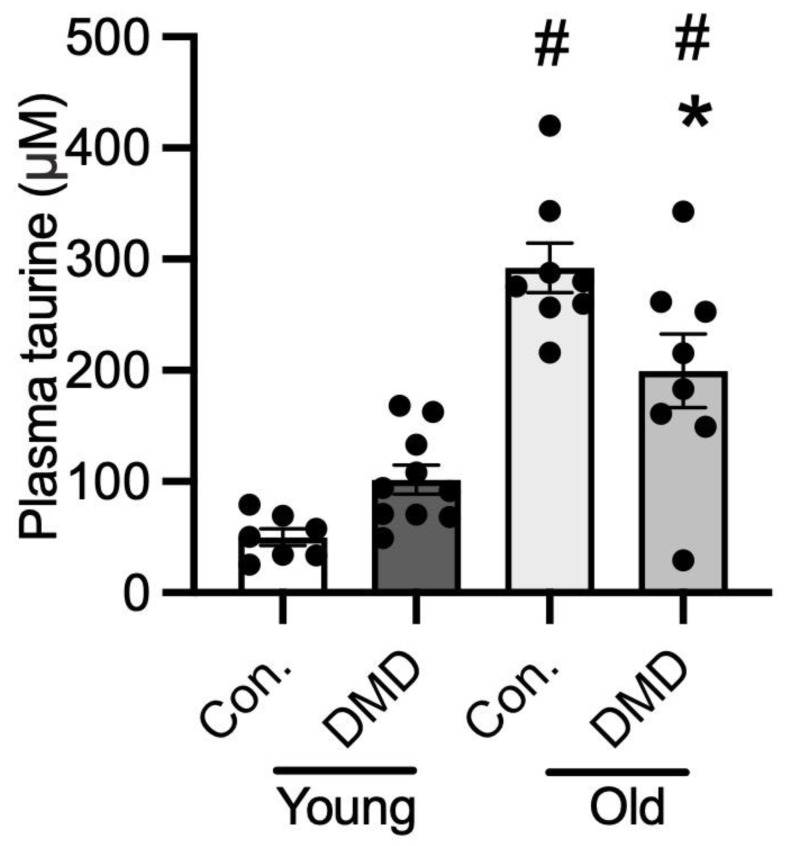
Human plasma levels of taurine in control males and patients with DMD. Compared for healthy normal aged-matched (A–M) controls and patients with DMD: ambulant (2–6 years) and non-ambulant (16–20 years). * = significantly (*p* < 0.05) different to healthy age-matched control. # = significantly (*p* < 0.05) different to young healthy age-matched control and ambulant DMD. Bars represent mean ± SEM and n = 8–10 per group.

**Table 1 metabolites-13-00232-t001:** Measures of phenotype and dystropathology in normal WT rats, compared with untreated and taurine-treated DMD^mdx^ rats at 12 weeks of age (data collected in France). * = DMD^mdx^ rats significantly (*p* < 0.05) different to WT control. $ = taurine treated DMD^mdx^ rats significantly (*p* < 0.05) different to untreated DMD^mdx^ rats. Values represent mean ± SEM and n = 8–10 per group. TA = tibialis anterior muscle, EDL = extensor digitorum longus muscle and Sol = soleus muscle.

Parameter	Wildtype	DMD^mdx^	DMD^mdx^ Taurine Treated- 2 g/kg/day	DMD^mdx^ Taurine Treated- 5 g/kg/day
Body weight (g)	497 ± 42	433 ± 35	438 ± 27	433 ± 33
TA weight (mg)	831 ± 12	952 ± 38 *	987 ± 29	1019 ± 71
EDL weight (mg)	221 ± 5	237 ± 9	241 ± 9	218 ± 13
Sol weight (mg)	207 ± 4	213 ± 7	214 ± 8	192 ± 12
Distance travelled (cm)	2305 ± 209	1937 ± 197	1885 ± 162	2012 ± 113
Activity time (s)	196 ± 9	181 ± 13	179 ± 10	191 ± 6
Number rearings	18 ± 2	11 ± 2	7 ± 2 *	12 ± 2
Platelets (Giga/L)	963 ± 35	1645 ± 52 *	1527 ± 37 *	1543 ± 50 *
Neutrophils (%)	11 ± 1	36 ± 6 *	36 ± 3 *	41 ± 5 *
Lymphocytes (%)	87 ± 2	60 ± 7 *	58 ± 3 *	54 ± 5 *
Hematocrit (%)	45 ± 1	48 ± 1 *	44 ± 0.4 $	45 ± 1$
Leucocytes (Giga/L)	8 ± 1	10 ± 1	10 ± 1	11 ± 1
Monocytes (%)	1.2 ± 1	3.7 ± 1	4.7 ± 0.5	3.5 ± 1

**Table 2 metabolites-13-00232-t002:** Taurine content of normal and DMD animal models and DMD patients at various ages. Models bolded are data from the current study. ↑ = increased taurine content compared to wildtype/normal control, ↓ = decreased, and — = no change in taurine content compared to wildtype/normal control, ✓ = improvement in dystropathology after taurine treatment, ✘ = no improvement in dystropathology after taurine treatment, N = not measured.

Model	Age	Taurine in Muscle	Taurine in Plasma	Taurine Efficacy	Reference
*mdx* mouse	18 days	—	↓	✓	[30]
	<3 weeks	↓	N	N	[32]
	23 days	—	N	✓	[21]
	4 weeks	↓/—	—	✓	[30]/[28]
	6 weeks	—	—	✓	[19,20,30,57]
	10 weeks	↑	—	✘	[28]
	12 weeks	↓	↑	N	[29]
	6 months	↓	N	✓	[27]
	6–8 months	—	↑	N	[24]
	12 months	↓	↑	Cardiac muscle only	[26]
**DMD^mdx^ rats**	12 weeks	↑	↑	✘	
** *dmd* ** **zebrafish**	6 dpf	↑ (tails)	N	✘	
GRMD dogs	8 months	↑	↓	N	[33]
**DMD patients**	2–6 years	N	—	N	
	16–20 years	N	↓	N	

## Data Availability

Not applicable.

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
