# Peer review of "Muscle Pathology in Dystrophic Rats and Zebrafish Is Unresponsive to Taurine Treatment, Compared to the *mdx* Mouse Model for Duchenne Muscular Dystrophy"

_metabolites, 2023, doi:10.3390/metabo13020232_

Round 1

Reviewer 1 Report

The authors of the paper entitled: Species differences in taurine metabolism and response to taurine treatment between dystrophic DMDmdx rat zebrafish and other models for Duchenne muscular dystrophy explore the use of taurine in many models of DMD” did a very interesting work in testing the use of taurine in other animal models. Despite the fact that there is no much novelty in using taurine in DMD, the authors study the metabolism and efficacy of taurine as adjuvant therapy even in less used models of DMD. It is an explorative study which opens new considerations in taurine metabolism in different species and different stages of the disease. Even though the study is well designed and the paper is solid. It would be interesting, as already stated by the authors, a natural history study in different species, together with treatments starting when the pathology is established. Another positive attitude in the paper is the detailed description of limitations of the study. However, there are some points that needs to be addressed:

Minor revisions:
- When you put the WT abbreviation for the first time (line111) it should be there that it goes in extenso.

- In the study 2 doses are used; please justify the choice of the 2 doses, and if well documented choose one or more pertinent references
- In the methods part, the authors describe various muscles (Biceps femoris for WB). When instead the authors quantify myeloperoxidase, it is not specified which muscle has been done on. Please specify.
- For what concern statistical analysis, please provide more details of what you mean for post hoc analysis.
- In Fig. 2 caption, please explain the abbreviation CD and CSD again
- One suggestion to the authors is that it would be probably better to see taurine’s action in a later stage of the pathology
- Something it would be better to rephrase is when the authors discuss taurine efficacy. From the literature it appears that the effect does not depends on taurine increment, but also from other factors. Probably this sentence (line 390) should be rephrased.

Author Response

Reviewer 1.

  1. When you put the WT abbreviation for the first time (line111) it should be there that it goes in extenso.

Response: This has been amended (now line 104 on revised document).

  1. In the study 2 doses are used; please justify the choice of the 2 doses, and if well documented choose one or more pertinent references.

Response: We now include the statement “These doses were chosen based on doses used in previous studies in our laboratory and others.” (lines 112-113)

  1. In the methods part, the authors describe various muscles (Biceps femoris for WB). When instead the authors quantify myeloperoxidase, it is not specified which muscle has been done on. Please specify.

Response: Biceps femoris was used for all muscle analysis, this has now been included in all sections.

  1. For what concern statistical analysis, please provide more details of what you mean for post hoc analysis.

Response: We corrected for multiple comparisons using Tukey post-hoc testing. This information has now been included in the statistical section (lines 247-248).

  1. In Fig. 2 caption, please explain the abbreviation CD and CSD again

Response: Done! 

  1. One suggestion to the authors is that it would be probably better to see taurine’s action in a later stage of the pathology.

Response: We appreciate the suggestion. The dystrophic rats were aged 3 months, whereas the zebrafish were very young: yet in mdx mice, taurine has been shown to be very beneficial at many ages (see Table 2), at least up to 6 months. In the context of the very young zebrafish, this includes very young mdx mice: for example, we showed striking prevention of the acute onset of myonecrosis with taurine administered to pre-weaned juvenile mice aged 14-22 days (ref 21- Terrill et al, 2016). An independent group administered taurine even earlier, in utero to mdxmice, with resultant similar striking benefits at 28 days; but far fewer benefits in older mice (aged 70 days) at a more stable phase of disease (ref 28- Barker et al, 2017). Thus, it seems unlikely that older dystrophic rats might be more responsive to the taurine treatment: instead there is the likelihood it will also be ineffective in older rats. 

  1. Something it would be better to rephrase is when the authors discuss taurine efficacy. From the literature it appears that the effect does not depends on taurine increment, but also from other factors. Probably this sentence (line 390) should be rephrased.

Response: We have rephrased this sentence plus we have also mentioned data from another study in mdx mice, to support the suggested scenario- “These previous results, combined with the new data in this current study, suggest that taurine is only efficacious in models where treatment improves muscle content of taurine. This increase after exogenous taurine treatment was not observed in dystrophic models where intrinsic endogenous muscle taurine levels were high (i.e DMDmdx rats, dmdpc2/pc2 zebrafish and 10 week mdx mice [28])” (lines 390-394).

Reviewer 2 Report

the manuscript entitled "Species differences in taurine metabolism and response to taurine treatment between dystrophic DMDmdx rat, zebrafish, and other models for Duchenne muscular dystrophy" nicely explained the effect of taurine in different species.

The following are some comments and suggestions that can be made to enhance the overall quality of this manuscript.

1. The title should be changed to be more precise and straightforward. ('other models' is not more appropriate here).

2. The introduction section appears to be too long; the first two paragraphs should be rewritten to be more precise and relevant.

3. content of Lines 96-101 can be moved to the discussion section.

4. section 2.6, "Muscle protein extraction and immunoblotting for taurine metabolites cysteine deoxygenase (CD) and cysteine sulfinate decarboxylase (CSD)" revise this heading as "Muscle protein extraction" is not well fit here.

5. Overall, the findings from the rat, zebrafish, and human samples are not comparable enough to draw any final conclusions about the effect of taurine and its metabolism.

Author Response

Reviewer 2.

  1. The title should be changed to be more precise and straightforward. ('other models' is not more appropriate here).

Response: Thank you for this suggestion. We have changed the title to read “Muscle pathology in dystrophic rats and zebrafish is unresponsive to taurine treatment, unlike the mdx mouse model for Duchenne muscular dystrophy”.  

  1. The introduction section appears to be too long; the first two paragraphs should be rewritten to be more precise and relevant.

Response: We made some minor edits to the Introduction to make it more concise, but we did not shorten extensively as we felt this material was important and relevant. 

  1. content of Lines 96-101 can be moved to the discussion section.

Response: We feel that a brief discussion of these results is important here as it explains why we performed the following experiment (human samples).

  1. "Muscle protein extraction and immunoblotting for taurine metabolites cysteine deoxygenase (CD) and cysteine sulfinate decarboxylase (CSD)" revise this heading as "Muscle protein extraction" is not well fit here.

Response: We have changed this heading to “Immunoblotting for taurine metabolites cysteine deoxygenase (CD) and cysteine sulfinate decarboxylase (CSD)”.

  1. Overall, the findings from the rat, zebrafish, and human samples are not comparable enough to draw any final conclusions about the effect of taurine and its metabolism.

Response: These studies indicate that the effects and metabolism of taurine differ across dystrophic species and the reasons for this remain to be clarified. We agree that additional studies are required before we can make conclusive statements about taurine metabolism in animal models for DMD, we have modified the text on page 14 (lines 464-468) to clarify this.

Reviewer 3 Report

This manuscript shows that taurine administration to the relatively novel animal models of DMD, the DMDmdx rat and the mdx zebrafish did not improve muscular function. The authors from Perth and other, mainly from Bari, earlier showed that taurine administration to mdx mice was effective on various parameters involving force.
The results have probably been disappointing to the authors. But it is important that they reach the public via this manuscript.
The latter is well-described, results are clear, details are given, the cited literature is complete (as far as I can tell), and the text is well-written. I appreciate Table 2 as it will help to stimulate further hypothesis on the yet incompletely understood species differences with respect to taurine and force.
The Discussion is in many ways a repetition of the results and it lacks the above-mentioned hypothesis why dystrophic mice react to taurine whereas rats don’t. It is worthwhile to speculate on that going beyond the differences in taurine levels in muscle at certain period of murine life.
In addition, the biochemical aspects of taurine action should be described: if it is purely an antioxidant, what part of the molecule gets oxidized and what in (which) protein is being reduced.
Therefore, I suggest that some parts that recapitulate Results are removed from the Discussion and that additional information regarding the two points mentioned is added to the Discussion.
Besides, I much appreciate the collaboration between the lab in Nantes and the two labs in Australia because the DMDmdx rat appears to have several important advantages over the mdx mouse and could perhaps replace the canine models (GRMD and the Tokyo-based “DMD”-beagle) that are time- and cost consuming and yield unsatisfactory results because of the large scatter.

Minor comments:
The uploaded pdf file shows most of them (3-fold, 8-week; use of adjective in e.g. “synthesis enzymes”), some commas, -D,L-serine, some words are lacking.
Line 366: Ref. 56 should be mentioned.
Line 429: there is a shift to the present form in “show”; check present/past in the manuscript.
Line 446: I propose to eliminate “in Japan” as it does not contribute to the assessment or understanding.

Author Response

Reviewer 3.

  1. The Discussion is in many ways a repetition of the results and it lacks the above-mentioned hypothesis why dystrophic mice react to taurine whereas rats don’t. It is worthwhile to speculate on that going beyond the differences in taurine levels in muscle at certain period of murine life. In addition, the biochemical aspects of taurine action should be described: if it is purely an antioxidant, what part of the molecule gets oxidized and what in (which) protein is being reduced. Therefore, I suggest that some parts that recapitulate Results are removed from the Discussion and that additional information regarding the two points mentioned is added to the Discussion

Response: We have addressed this point at the beginning and throughout the Discussion by removing some text that reiterated results, and have emphasised our hypothesis, and the action of taurine (with substantial text changes in yellow highlight).

  1. Minor comments:
    The uploaded pdf file shows most of them (3-fold, 8-week; use of adjective in e.g. “synthesis enzymes”), some commas, -D,L-serine, some words are lacking. 
    Line 366: Ref. 56 should be mentioned. 
    Line 429: there is a shift to the present form in “show”; check present/past in the manuscript.
    Line 446: I propose to eliminate “in Japan” as it does not contribute to the assessment or understanding

Response: Thank you for pointing out these errors, they have been amended.

Reviewer 4 Report

The manuscript entitled "Species differences in taurine metabolism and response to taurine treatment between dystrophic DMDmdx rat, zebrafish, and other models for Duchenne muscular dystrophy" by Terrill et al. indicated relationship between taurine metabolism and DMD pathology. This study is very imporatnant and interesting. But, some correction may be needed for publication. In Abstract, it is better to clear and highlight research questions in this study. In Introduction section, it is better to add previous findings and backgrounds of taurine and DMD, and show Gap of conventional tratment methods of DMD.  In discussion section, it is better to add advantage(s)/limitation(s) of this study. As for evaluation of DMD pathology, it has been used to analyze the serum levels of microRNAs in DMD patients and models. Therefore, it is better to add microRNAs levels in this study.  

Author Response

Reviewer 4.

  1.  In Abstract, it is better to clear and highlight research questions in this study.

Response: We have amended the abstract to more clearly state our research objectives.

  1. In Introduction section, it is better to add previous findings and backgrounds of taurine and DMD, and show Gap of conventional tratment methods of DMD. 

Response: This Introduction is already quite extensive, and we aimed to keep it concise,   especially since Reviewer #2 (point 9) suggested it could  be shortened (in direct contrast with Reviewer 4). Thus, we did not expand it further.

  1. In discussion section, it is better to add advantage(s)/limitation(s) of this study.

Response: We have inserted text related to advantages and limitations into the Discussion at the end (from line 466).                                                                                

  1. As for evaluation of DMD pathology, it has been used to analyze the serum levels of microRNAs in DMD patients and models. Therefore, it is better to add microRNAs levels in this study.  

Response: We are aware of the value of microRNAs as a potential biomarker of dystropathology (ref 8, Grounds et al, 2020). However, since all of the other validated biomarkers of pathology we analysed did not change, we did not see the value in additional readouts.